# Chemical and Pharmacological Screening of *Rhinella icterica* (Spix 1824) Toad Parotoid Secretion in Avian Preparations

**DOI:** 10.3390/toxins12060396

**Published:** 2020-06-15

**Authors:** Raquel Soares Oliveira, Bruna Trindade Borges, Allan Pinto Leal, Manuela Merlin Lailowski, Karla de Castro Figueiredo Bordon, Velci Queiróz de Souza, Lúcia Vinadé, Tiago Gomes dos Santos, Stephen Hyslop, Sidnei Moura, Eliane Candiani Arantes, Alexandre Pinto Corrado, Cháriston A. Dal Belo

**Affiliations:** 1Laboratório de Neurobiologia e Toxinologia, Programa de Pós-Graduação em Ciências Biológicas (PPGCB), Universidade Federal do Pampa (UNIPAMPA), Avenida Antônio Trilha 1847, São Gabriel RS 97300-000, Brazil; raquelsoaresoliveira@yahoo.com.br (R.S.O.); brunaborges@alunos.unipampa.edu.br (B.T.B.); allan-leal@hotmail.com (A.P.L.); velcisouza@unipampa.edu.br (V.Q.d.S.); 2Programa de Pós-Graduação em Ciências Biológicas: Bioquímica Toxicológica (PPGBTox), Universidade Federal de Santa Maria (UFSM), Avenida Roraima 1000, Santa Maria RS 97105-900, Brazil; 3Laboratório de Biotecnologia de Produtos Naturais e Sintéticos, Instituto de Biotecnologia, Universidade de Caxias do Sul (UCS), Rua Francisco Getúlio Vargas 1130, Caxias do Sul RS 95070-560, Brazil; manusnowwhite@yahoo.com.br (M.M.L.); smsilva11@ucs.br (S.M.); 4Departamento de Ciências BioMoleculares, Faculdade de Ciências Farmacêuticas de Ribeirão Preto, Universidade de São Paulo (USP), Avenida do Café, s/n, Ribeirão Preto SP 14.040-903, Brazil; karla@fcfrp.usp.br (K.d.C.F.B.); ecabraga@fcfrp.usp.br (E.C.A.); 5Laboratório de Estudos em Biodiversidade Pampiana, Universidade Federal do Pampa (UNIPAMPA), Avenida Antônio Trilha 1847, São Gabriel RS 97300-000, Brazil; frogomes@gmail.com; 6Departamento de Farmacologia, Faculdade de Ciências Médicas, Universidade Estadual de Campinas (UNICAMP), Rua Tessália Vieira de Camargo, 126, Cidade Universitária Zeferino Vaz, Campinas SP 13083-887, Brazil; hyslop@unicamp.br; 7Departamento de Farmacologia, Faculdade de Medicina de Ribeirão Preto, Universidade de São Paulo (USP), Avenida Bandeirantes 3900, Ribeirão Preto SP 14040-030, Brazil; apcorrad@fmrp.usp.br

**Keywords:** Anti-AChE activity, avian, chick neurobiological preparations, cytotoxicity, neuromuscular blockade, toad poison

## Abstract

The biological activity of *Rhinella icterica* parotoid secretion (RIPS) and some of its chromatographic fractions (RI18, RI19, RI23, and RI24) was evaluated in the current study. Mass spectrometry of these fractions indicated the presence of sarmentogenin, argentinogenin, (5*β*,12*β*)-12,14-dihydroxy-11-oxobufa-3,20,22-trienolide, marinobufagin, bufogenin B, 11α,19-dihydroxy-telocinobufagin, bufotalin, monohydroxylbufotalin, 19-oxo-cinobufagin, 3α,12*β*,25,26-tetrahydroxy-7-oxo-5*β*-cholestane-26-*O*-sulfate, and cinobufagin-3-hemisuberate that were identified as alkaloid and steroid compounds, in addition to marinoic acid and *N*-methyl-5-hydroxy-tryptamine. In chick brain slices, all fractions caused a slight decrease in cell viability, as also seen with the highest concentration of RIPS tested. In chick *biventer cervicis* neuromuscular preparations, RIPS and all four fractions significantly inhibited junctional acetylcholinesterase (AChE) activity. In this preparation, only fraction RI23 completely mimicked the pharmacological profile of RIPS, which included a transient facilitation in the amplitude of muscle twitches followed by progressive and complete neuromuscular blockade. Mass spectrometric analysis showed that RI23 consisted predominantly of bufogenins, a class of steroidal compounds known for their cardiotonic activity mediated by a digoxin- or ouabain-like action and the blockade of voltage-dependent L-type calcium channels. These findings indicate that the pharmacological activities of RI23 (and RIPS) are probably mediated by: (1) inhibition of AChE activity that increases the junctional content of Ach; (2) inhibition of neuronal Na^+^/K^+^-ATPase, leading to facilitation followed by neuromuscular blockade; and (3) blockade of voltage-dependent Ca^2+^ channels, leading to stabilization of the motor endplate membrane.

## 1. Introduction

Brazil is the country with the richest biodiversity of anuran amphibians, with 1093 catalogued species, of which at least 42 are on the list of endangered species [1,2,3]. Of particular interest are the biological activities of anuran gland and skin secretions, not only because of their physiological, behavioral, and ecological functions, but also because their complex composition provides a rich source of novel chemicals with potential biotechnological and therapeutic applications [4,5,6,7,8]. In addition, many anurans (and amphibians in general) have complex mechanisms to avoid diseases, parasites, and predation that frequently involve the use of a variety of secretions, the composition of which may be modulated by environmental factors [9,10,11].

Toads possess cutaneous glands that are broadly classified into two types: mucous glands, that produce mucus involved in cutaneous respiration, thermoregulation, and reproduction; and granular glands, that secrete noxious or toxic compounds used as a defense against predators [12,13,14]. These secretions consist mainly of alkaloids, steroids (bufadienolides and bufotoxins), biogenic amines (epinephrine, norepinephrine, serotonin, bufotenine, and dehydrobufotenine), proteins, and peptides [6,12,13,15], and act by direct contact with the mucous membranes of the mouth, eye, or throat of potential predators [9,16].

In traditional Chinese, Indian, and Egyptian medicine, cardiotonic steroids from plants, animals, or minerals have been widely used for millennia to treat sores, heart failure, pain, and different types of cancer [17,18,19,20]. For therapeutic applications, the most important amphibian secretions are obtained from toads of the family Bufonidae [6,21] and have been widely used in America, Asia, and Europe as antiviral agents (to treat human immunodeficiency virus (HIV)), as well as for their anti-proliferative [22,23,24], antibacterial [25], antiparasitic [26], insecticidal [27,28], antidiabetic [29], anti-cancer [30,31], anti-inflammatory, and analgesic [7,32,33,34] activities. Many compounds that occur in toad secretions, e.g., bufalin, telocinobufagin, hellebrin, marinobufagin, and cinobufagin, can vary markedly among individuals, geographic regions, and species in response to environmental conditions, e.g., temperature, and dietary composition, and may involve morphological adaptations [14,35,36].

*Rhinella icterica* (Spix, 1824), the “yellow cururu toad”, is a large anuran of the family Bufonidae that is native to South America and occurs in Brazil, Paraguay, and Argentina [3,37]. *Rhinella icterica* parotoid secretion (RIPS) has antimicrobial activity against *Escherichia coli* and *Staphylococcus aureus* [38]. In a previous report, we showed that RIPS was toxic to the peripheral and central nervous systems of avian, mammalian, and insect species through a digitalis-like action [27,39]. Although preliminary studies of RIPS have identified some chemical components, such as dehydrobufotenine, hellebrigenin, telocinobufagin, marinobufagin, bufotalin, bufalin, and 19-oxo-cinobufagin [13,36,39], no previous study has sought to examine the chemical constitution of this secretion in detail and to assess its specific interaction with the avian nervous system. In this work, we investigated the chemical composition of RIPS to identify the main groups of compounds responsible for the acute neurotoxicity caused by this secretion at avian skeletal neuromuscular junctions.

Based on pharmacological screening in tissue slices and avian neuromuscular preparations, the most important finding of this work was that the two main neurotoxic effects (muscle twitch-tension facilitation and neuromuscular blockade) observed in *Rhinella icterica* fraction 23 (RI23) could be dissociated; this observation suggested the presence of at least two different molecules responsible for the overall toxicity of the secretion. The action of RI23 at the avian neuromuscular junction was complex and involved an increase in acetylcholine (ACh) release via an ouabain-like mechanism combined with anti-acetylcholinesterase (anti-AChE) activity and a membrane-stabilizing activity similar to propranolol.

## 2. Results

### 2.1. Biochemical Characterization of RIPS by FPLC

Fractionation of RIPS (5 mL of a 2 mg/mL solution) by reversed-phase fast protein liquid chromatography (RP-FPLC) on a C18 column resulted in 30 fractions (Figure 1a), all of which were initially screened for neuromuscular activity in chick *biventer cervicis* preparations. Only fractions 18, 19, 23, and 24 showed neuromuscular activity and these were therefore chosen for further investigation. The electrophoretic profile of RIPS showed the presence of proteins ranging from ~10 kDa to 200 kDa, with the main components around 45 kDa, 97 kDa, and 200 kDa (Figure 1b). Fractions RI18, RI19, RI23, and RI24 showed no proteins below 200 kDa.

Screening of these fractions in chick *biventer cervicis* preparations showed that only RI23 reproduced the activity of RIPS by facilitating the twitch-tension responses prior to causing irreversible neuromuscular blockade. In contrast, fractions RI18, RI19, and RI24 caused only neuromuscular blockade. Analysis of these four peaks by high resolution mass spectrometry (HRMS) with electrospray ionization (ESI) in both modes revealed the presence of the following compounds: 3α,12*β*,25,26-tetrahydroxy-7-oxo-5*β*-cholestane-26-O-sulfate, (5*β*,12*β*)-12,14-dihydroxy-11-oxobufa-3,20,22-trienolide, 11α,19-di-hydroxy-telocinobufagin, 19-oxo-cinobufagin, argentinogenin, bufogenin B, bufotalin, cinobufagin-3-hemisuberate, marinobufagin, marinoic acid, monohydroxylbufotalin, *N*-methyl-5-hydroxytryptamine, and sarmentogenin (Table 1, Figure 2).

### 2.2. Neuromuscular Blockade Caused by RIPS and Its Fractions in Chick Biventer Cervicis Preparations

RIPS and fractions RI18, RI19, RI23, and RI24 were screened for neuromuscular activity in chick *biventer cervicis* preparations at a single concentration of 10 µg/mL (Figure 3). RIPS caused marked facilitation of the twitch-tension responses (52 ± 6% increase, *n* = 3; *p* < 0.05 compared to saline control preparations) in field-stimulated preparations that was maximal at ~20 min after the addition of secretion to the organ and was followed by a progressive decrease in muscle tension until complete abolition of neuromuscular responses after 110–120 min (Figure 3). In contrast to RIPS, RI18 (Figure 3a) and RI19 (Figure 3b) caused only neuromuscular blockade without any prior facilitation. The progressive decrease in muscle twitch-tension caused by these two fractions reflected that seen with RIPS, but was more rapid, with RI18 being more potent (caused quicker blockade) than RI19 (total blockade within 30 min and 40 min, respectively). RI23 mimicked the response to RIPS in that it caused muscle facilitation that was maximal after 10 min (32 ± 9% increase in tension, *p* < 0.05, *n* = 3) followed by rapid abolition of twitch-tension responses (total blockade by 20 min after addition to the organ bath compared to 110–120 min for RIPS; Figure 3c). The neuromuscular activity of RI24 differed from the other fractions in that it did not cause muscle facilitation and the blockade it produced was slower, started soon after addition to the organ bath and was incomplete (maximal blockade of 92 ± 8% after 110–120 min, *n* = 3; Figure 3d); thus, the time-course of the blockade differed completely from that seen with RIPS and RI18, RI19, and RI23. In this same preparation, 0.4 nM ouabain, a specific inhibitor of Na^+^/K^+^-ATPase (NKA), produced a combination of responses that was identical to that seen with RI23 (Figure 3e) and similar to that caused by RIPS. Specifically, there was a transient facilitatory response (50 ± 5%, *n* = 3) that was maximal after 10 min and greater than that seen with RI23 (see above), followed by complete irreversible neuromuscular blockade within 30 min.

RIPS and the four fractions had variable effects on the *biventer cervicis* muscle contractures to exogenous ACh and KCl. RI18 and RI23 attenuated the responses to ACh and KCl, with RI23 abolishing the contracture to exogenous ACh and virtually abolishing the response to KCl (Figure 3f). RI24 reduced the responses only to KCl and RI19 had no significant effect on contractures to either agonist, despite a 31 ± 15% decrease in the response to KCl (Figure 3f). The contractures to KCl were more consistently affected by RIPS and fractions RI18, RI23, and RI24 than the responses to ACh (reductions of 73 ± 15%, 61 ± 10%, 91 ± 5%, and 59 ± 9% for RIPS, RI18, RI23, and RI24, respectively; *p* < 0.05, *n* = 3). Under similar conditions, ouabain had no effect on ACh-induced contractures but reduced the responses to KCl by 66 ± 4% (*n* = 3); this profile was similar to that seen with RI24 (Figure 3f).

### 2.3. Effect of RIPS Fractions on Acetylcholinesterase (AChE) Activity

In view of the ability of RIPS and RI23 to cause muscle facilitation prior to blockade, a finding suggestive of ACh accumulation in the skeletal neuromuscular junction, we examined the ability of RIPS and its fractions to inhibit AChE activity in homogenates of chick *biventer cervicis* preparations that had been incubated with RIPS, RI18, RI19, RI23, or RI24 (10 µg/mL each) or neostigmine (5 µM, a classic AChE inhibitor) during the myographic experiments. All treatments significantly inhibited the AChE activity of the homogenates (Figure 4a). In contrast, the incubation of brain slices with RIPS and its fractions (5 µg/mL each) for 1 h had no marked effect on AChE activity compared to the negative control (HEPES (4-(2-hydroxyethyl)piperazine-1-ethanesulfonic acid) buffer), whereas neostigmine (5 µM, positive control) strongly inhibited this activity (Figure 4b).

### 2.4. Effect of RIPS on Propranolol-Induced Neuromuscular Blockade

In view of the ability of RIPS to inhibit AChE (see above) and its potential inhibition of Na^+^/K^+^-ATPase (based on the identical profiles of RI23 and ouabain in chick *biventer cervicis* preparations), we compared the effects of RIPS (10 µg/mL) with those of the non-selective β-blocker propranolol in chick *biventer cervicis* preparations since propranolol has a similar profile of activity (inhibition of AChE and Na^+^/K^+^-ATPase activities) to that of RIPS. Incubation of chick *biventer cervicis* preparations with propranolol (10 µg/mL) caused a progressive decrease in the twitch-tension after the addition to the organ bath, with a maximum reduction of 43 ± 8% after 120 min (Figure 5a). When RIPS was added 60 min after propranolol, it still produced its characteristic response, but of smaller magnitude and a shorter time scale (complete blockade within 50 min of RIPS addition, compared to 110–120 min with RIPS alone; Figure 5a). When the order was reversed, i.e., RIPS added before propranolol, the latter had no major effect on the overall shape of the response to RIPS except to shift the curve slightly to the left, indicating a somewhat faster blockade (50% blockade without propranolol in 43 ± 5 min compared to 32 ± 4 min, but the difference was not significant; complete blockade also occurred earlier: after 72 ± 7 min compared to 110–120 min with RIPS alone, *p* < 0.01, *n* = 4; Figure 5b).

In experiments to examine the effect on the responses to exogenous ACh and KCl, propranolol (10 µg/mL) markedly attenuated the ACh-induced contractures but did not affect those to KCl. In protocols where propranolol was added before RIPS, this pattern was maintained but more intensely, i.e., greater effect. In contrast, when the order of addition was reversed (RIPS before propranolol) this distinctive pattern was abolished and instead there was a slight but non-significant attenuation in the responses to both ACh and KCl (Figure 5c).

### 2.5. Cell Viability in the Presence of RIPS and Its Fractions

We have previously shown that RIPS can partially prevent excitotoxicity in mouse brain slices [39]. To confirm this anti-neurotoxic potential, chick brain slices were incubated with RIPS (5, 10, 20, and 40 µg/mL) and its four fractions (RI18, RI19, RI23, and RI24; 5 µg/mL each). For RIPS, only the highest concentration tested (40 µg/mL) caused a small decrease (14 ± 4%) in cell viability (Figure 6a). For comparison, digoxin, a known Na^+^/K^+^-ATPase inhibitor, also decreased the viability of chick brain slices by 35 ± 6% at the highest concentration tested (40 µg/mL; Figure 6b). Since RIPS did not adversely affect cell viability in chick brain slices at concentrations up to 5 µg/mL, we chose this concentration to screen the RIPS fractions for their effect on cell viability. All subfractions caused a slight but significant decrease in cell viability that ranged from 13 ± 4% (RI24) to 23 ± 2% (RI19) compared to the negative control (brain slices incubated with HEPES buffer alone). The positive control (hydrogen peroxide, H_2_O_2_) reduced cell viability by ≥90% in all cases (Figure 6c).

## 3. Discussion

In previous work, we examined the effects of RIPS in insect neurological and neuromuscular preparations [27,39] and in rat cardiac tissue [39]. As a continuation of these investigations, in this study we investigated the activity of RIPS and some of its chromatographic fractions in avian central and peripheral nervous tissue and undertook chemical screening to identify the main compounds associated with the pharmacological activity of RIPS. RP-FPLC of the secretion resulted in four fractions (RI18, RI19, RI23, and RI24) that, together, accounted for the neurobiological activity of RIPS. Analysis of the four fractions by liquid chromatography-diode array detection-mass spectrometry (LC-DAD-MS) identified a variety of compounds (see Table 1) broadly classified as alkaloids and steroids, in addition to marinoic acid and *N*-methyl-5-hydroxytryptamine [43,59].

Of the four fractions, only RI23 completely mimicked the activity of RIPS in chick *biventer cervicis* preparations in which it produced a biphasic response involving a transitory facilitation of twitch-tension followed by progressive neuromuscular blockade (Figure 3c). The occurrence of facilitation suggested an increase in ACh release and/or a lack of degradation of this neurotransmitter at the motor endplate [61,62].

We have recently shown that RIPS inhibits Na^+^/K^+^-ATPase activity in rat cardiomyocytes in a manner similar to ouabain [39]. At vertebrate neuromuscular junctions, the inhibition of Na^+^/K^+^-ATPase increases ACh release by inducing depolarization of the motor somatic nerve terminal [62] and can lead to muscle facilitation, as also seen here (Figure 3e). In our experimental conditions, the inhibition of AChE activity by RIPS probably also prevented ACh degradation at chick neuromuscular junctions. This latter pharmacological activity is also shared with propranolol, a non-selective β-blocker [63,64]. Indeed, as shown here, propranolol intensified the neuromuscular blocking activity of RIPS (Figure 5b). This activity of propranolol may be related to its stabilizing effect on the motor endplate membrane, more commonly known as a “local anesthetic” action [63], and suggests that it may share a similar molecular target with RIPS.

The chemical analysis of RI23 indicated a predominance of bufogenins, which are steroidal compounds from toad parotoid secretions that exert a cardiotonic effect via a digoxin- or ouabain-like mechanism [9,34] and the blockade of voltage-dependent L-type calcium channels; one of the best studied bufogenins is bufalin [65]. The findings of our investigation indicate that the pharmacological activity of RI23 at the avian motor endplate involves at least three mechanisms, namely, (1) anti-AChE activity; (2) the inhibition of Na^+^/K^+^-ATPase activity that could contribute to facilitation and subsequent neuromuscular blockade; and (3) the blockade of voltage-dependent Ca^2+^ channels, possibly mediated by stabilization of the motor endplate membrane.

Screening of the other three fractions (RI18, RI19, and RI24) in chick *biventer cervicis* muscle preparations showed that they caused only progressive neuromuscular blockade, with no initial facilitation. Chemical analysis of these fractions revealed the presence of serotonin [40], the bufadienolide-related marinoic acid [59] and the cardiac glycoside-analog bufotalin [57]. The lack of a facilitatory response with these fractions could reflect differences in their chemical composition compared to RI23 or could simply be because they had a lower content of the compound that caused facilitation. Since we did not systematically screen the fractions for their biological activities based on the proportion in which the fractions occurred in RIPS, this question remains unanswered, although the relative order of contribution would likely be similar to that seen here. A further limitation was the fact that we did not investigate the responses to possible combinations of these fractions.

The skin secretion of *Rhinella* toads contains significant amounts of bufadienolides that are potent cardiac glycosides, as well as catecholamines and indolylalkylamines [9,40]. The last of these groups includes *N*-methyl analogs of serotonin (*N*-methyl-5-hydroxytryptamine). Patten et al. [66] reported serotonin-induced depression of isometric twitches in rat skeletal muscle, and Meltzer et al. [67] suggested a direct effect of serotonin on rat skeletal muscles, independent of vascular changes. At skeletal neuromuscular junctions, serotonin may function as an endogenous allosteric modulator of nicotinic acetylcholine receptors (nAChR) [68].

The other components in fractions RI18 and RI19 may intensify the decrease in muscle contractility caused by RIPS, possibly via a propranolol-like stabilizing effect on the postsynaptic membrane [69]. In this regard, it is worth noting that propranolol affected almost exclusively the responses to exogenous ACh, a receptor-mediated phenomenon (that would be most affected by membrane stabilization), compared to the essentially ion-mediated effect of high potassium. Propranolol also slightly potentiated the attenuation of muscle contractility mediated by RIPS in chick neuromuscular preparations (Figure 5b). The observation that when given after RIPS propranolol had little effect on the response to exogenous ACh could reflect the anti-AChE activity of RIPS that would increase the junctional content of ACh and prevent the binding of propranolol to the nAChR [70].

RIPS and its fractions markedly inhibited muscle AChE activity. Since AChE degrades ACh and terminates the pharmacological activity of this neurotransmitter at cholinergic synapses, we suggest that the anti-AChE activity of RIPS contributes to the transitory facilitation in chick *biventer cervicis* preparations [70]. The anti-AChE activity of RIPS may also contribute to the progressive neuromuscular blockade by inducing ACh-dependent hyperpolarization of skeletal muscle fibers [71]. The inability of RIPS to inhibit avian brain AChE compared to the muscle enzyme may reflect the insensitivity of this AChE to the secretion, in a manner analogous to the insensitivity of brain AChE to inhibition by fasciculins from mamba snake venoms [72].

As indicated here, RIPS and its fractions showed no neuroprotective activity in the avian central nervous system (chick brain slices); indeed, there was a slight decrease in cell viability, as also seen with digoxin. The four fractions tested showed a slightly greater ability to reduce cell viability than RIPS. This discrepancy may reflect the fact that the fractions were screened in a “purer” state than RIPS and most likely therefore contained a greater amount of active components than unfractionated secretion (a similar argument could also explain the greater effect of fractions RI18, RI19, and RI23 in causing neuromuscular blockade compared to RIPS). Alternatively, the presence of protective (inhibitory) factors in the whole secretion could have attenuated the action of the components of fractions RI18, RI19, RI23, and RI24. During chromatography, these protective factors would be separated from the four fractions, allowing the latter to reduce cell viability to a greater extent than seen with RIPS. Cell viability can be altered by toxic conditions that include the production of reactive oxygen species, excitotoxicity, necrosis and apoptosis [73,74,75,76]. In addition, anti-AChE compounds can protect cells from neuronal death and cognitive impairment in neurodegenerative diseases [31,33,76,77,78,79,80]. The inhibition of Na^+^/K^+^-ATPase is one of the mechanisms implicated in the cytotoxicity of cardiac glycosides [81]. In Na^+^/K^+^-ATPase inhibition, the resulting elevation in the intracellular concentration of Na^+^ would compromise mitochondrial energetics and the redox balance by blunting the mitochondrial accumulation of Ca^2+^, thereby contributing to the possible cytotoxicity of cardiac glycosides.

In terms of biotechnological applicability, the facilitatory action of fraction RI23 could indicate a potential use in dystrophic neuromuscular diseases, such as Duchenne and myasthenic syndromes, and as a memory enhancer to treat central nervous system dysfunctions, e.g., Alzheimer´s disease, as also suggested for crotamine, a low molecular mass (~4.9 kDa) basic peptide from *Crotalus durissus terrificus* (South American rattlesnake) venom [82,83].

## 4. Conclusions

This is the first report to evaluate the effect of RIPS and some of its components in avian central and peripheral neurobiological preparations. The chemical analysis of RIPS revealed the presence of four main fractions (RI18, RI19, RI23, and RI24) that contained at least 13 different compounds, most of which shared chemical identities with steroids and indolylalkylamines. Fraction RI23 reproduced the entire pharmacological profile of the venom and contained bufadienolides and cardiac glycosides. The pharmacological activity of RI23 in the avian peripheral nervous system was complex and reflected interactions with several molecular targets, including AChE, the Na^+^/K^+^-ATPase pump and L-type Ca^2+^ channels, as well as stabilization of the cell membrane. The similarities of the pharmacological profile of RIPS with that of propranolol suggested a novel pharmacological function for the toxins of this secretion. The results of this investigation expand our understanding of the biological activities of RIPS and reinforce the biotechnological potential of its individual compounds.

## 5. Materials and Methods

### 5.1. Reagents and Secretion Collection

All chemicals and reagents used were of the highest purity and were obtained from Sigma–Aldrich (St. Louis, MO, USA), Merck (Rio de Janeiro, RJ, Brazil), or BioRad Laboratories (Hercules, CA, USA). The parotoid secretion was collected as previously described [39] using toads captured near the city of Derrubadas, in the southern Brazilian state of Rio Grande do Sul. The toad capture and secretion collection were authorized by the System of Authorization and Information in Biodiversity (SISBIO, permit n^o^. 24867-2). The toad secretion was collected by manual compression of the large post-orbital parotoid glands of adult male and female toads, essentially as described by Leal et al. [27]. The yellowish, viscous secretion was weighed (in liquid form) on a Shimadzu high precision analytical balance. Two grams of liquid RIPS was treated with 50 mL of methanol for three days at room temperature followed by lyophilization (Liobras, Liotop K105, São Paulo, Brazil) and yielded a powdered extract, as described by Rostelato-Ferreira et al. [84]. This extract was subsequently dispersed in 0.1% trifluoroacetic acid (TFA) at a concentration of 2 mg/mL, sonicated for 10 min, centrifuged (5000× *g*, 5 min) and the supernatant was dried, named *Rhinella icterica* parotoid secretion (RIPS) and used to perform the assays.

### 5.2. Animals

Hyline chicks (1–10 days) were obtained from a local supplier (Agropecuaria Sinuelo, São Gabriel, RS, Brazil). The chicks were housed 15 per cage with water and food ad libitum at 25 °C on a 12-h light/dark cycle. They were acclimatized for two days prior to use in the experiments. This work was approved by an Institutional Committee for Ethics in Animal Use (CEUA/UNIPAMPA, protocol n°. 028/2019) and the experiments were done according to the general ethical guidelines for animal use established by the Brazilian Society of Laboratory Animal Science (SBCAL) and Brazilian legislation (Federal Law n^o^. 11,794, of 8 October 2008), in conjunction with the guidelines for animal experiments established by the Brazilian National Council for the Control of Animal Experimentation (CONCEA).

### 5.3. RP-FPLC of Parotoid Secretion

In each chromatographic run, 5 mL of a solution of RIPS (2 mg/mL) was applied to a reversed-phase C18 column (5 μm, 250 × 10.0 mm, 300 Å; Jupiter^®^, Phenomenex) coupled to an Äkta Pure fast protein liquid chromatography (FPLC) system (GE Healthcare, Piscataway, NJ, USA), as described elsewhere [85]. The chromatographic system was operated by Unicorn 5.20 software (General Electric (GE) Healthcare). The column was initially equilibrated with 0.1% TFA (solution A) and the fractions then eluted with a step-wise gradient (0-50% and 50-100%) of acetonitrile (60% acetonitrile in 0.1% TFA; solution B) until 100% of solution B was reached. The column was eluted at a flow rate of 1 mL/min and the elution profile was monitored at 214 nm. The resulting fractions were collected and stored at −20 °C until tested. An aliquot (2 µg) of each fraction was dried and resuspended in ultrapure water prior to Tricine-SDS-PAGE [86]). A sample (60 μg) of RIPS used for RP-FPLC was also run simultaneously in the same gels as the chromatographic fractions. Ultra-low range molecular mass markers (1060–26,600 Da, Sigma–Aldrich M3546) and wide-range molecular mass markers (6500–200,000 Da, Sigma–Aldrich S8445) were used to estimate the molecular masses of the RIPS components. The gels were run at 85 V, 33 mA, and 3 W for 4 h and then stained with silver nitrate [87].

### 5.4. High Resolution Mass Spectrometry

For mass spectrometry, the dried fractions were dissolved in a solution consisting of 50% (*v*/*v*) chromatographic grade acetonitrile (Tedia, Fairfild, OH, USA) and 50% (*v*/*v*) deionized water, to which 0.1% formic acid and 0.1% ammonium formate had been added, for analysis in positive and negative electrospray ionization modes (ESI(+) and ESI(−), respectively). The individual solutions were infused directly into the ESI source via a syringe pump (Harvard Apparatus, Hamilton, Reno, NV, USA), at a flow rate of 180 μL/min. The ESI(+)-and ESI(−) mass spectrometric (MS) and tandem MS-MS profiles were acquired using a hybrid high-resolution and high-accuracy (5 μL/L) micrOTOF-Q mass spectrometer (Bruker Scientific^®^, Billerica, MA, USA) under the following conditions: capillary and cone voltages were set to +3500 and +40 V, respectively, with a desolvation temperature of 200 °C. The collision-induced dissociation energy (CID) for the ESI(+) MS-MS was optimized for each component. The diagnostic ions were identified by comparison of their dissociation patterns, exact mass and isotopic ratio, with compounds identified in previous studies. For data acquisition and processing, time-of-flight (TOF) control and data analysis software (Bruker Scientific^®^) was used. The data were collected in the 70–1200 *m*/*z* range, at a rate of two scans/s, providing 50,000 full width at half maximum (FWHM) resolution at 200 *m*/*z*. No important ions were observed below 90 *m*/*z* and above 950 *m*/*z*, so the ESI(+)-MS data are shown for the range of 90–950 *m*/*z*.

### 5.5. Biological Assays

#### 5.5.1. Tissue Slice Preparation and Treatment

The chicks were euthanized by decapitation under anesthesia with halothane and the brain was dissected and placed in HEPES ((4-(2-hydroxyethyl)piperazine-1-ethanesulfonic acid))-saline buffer (124 mM NaCl, 4 mM KCl, 1.2 mM MgSO_4_, 12 mM glucose, 1 mM CaCl_2_, and 25 mM HEPES, pH 7.4) at 4 °C. The solution was previously oxygenated for 30 min. The cortical region of the brain was separated and slices 400 μm thick were prepared with a McIlwain tissue slicer [88]. The diameter of the slices was standardized using 3 mm circular tissue cutters or punches. The tissue slices were subsequently transferred to 96-well plates containing HEPES–saline buffer (200 μL/slice). After a 30 min pre-incubation, the buffer was replaced with control solutions (HEPES–saline as the negative control and 44 mM H_2_O_2_ as the positive control) or fractions (RI18, RI19, RI23, RI24) and incubated at 37 °C for 60 min. All fractions were dissolved in HEPES–saline buffer and were tested at a fixed concentration of 5 μg/mL. The concentrations used here were based on previous work by our group which showed that RIPS caused maximum neuromuscular interference, AChE inhibition and a reduction in cell viability at concentrations of 5–10 µg/mL [39].

#### 5.5.2. Cell Viability

The colorimetric MTT (3-(4, 5-dimethylthiazol-2-yl)-2, 5-diphenyltetrazolium bromide) reduction assay based on cellular oxidative metabolism was used to assess cell viability [73]. The assay was done using chick brain slices in the absence or presence of RIPS fractions. Immediately after incubation as described above, the slices were placed in HEPES–saline containing 0.05% MTT at 37 °C for 30 min. During this period, the MTT was converted into a purple formazan product after cleavage of the tetrazolium ring by dehydrogenases. The formazan was subsequently dissolved by adding 100% dimethyl sulfoxide (DMSO), resulting in a colored compound that could be monitored by reading the absorbance at 490 nm in a Biotek ELx800 multi-well plate reader (Biotek Instruments, Winooski, VT, USA); the results were analyzed with Gen5 Data Analysis software (Biotek Instruments).

#### 5.5.3. Chick Biventer Cervicis Preparation

Chick *biventer cervicis* neuromuscular preparations were used to characterize the neuromuscular activity of RIPS fractions. The chicks were killed with an overdose of anesthetic (halothane) and the *biventer cervicis* muscle and associated nerve were isolated and mounted as described by Ginsborg and Warriner [89]. The muscles were mounted (tension: 0.5 g/cm) in 5 mL isolated organ baths (AVS Projetos, São Paulo, SP, Brazil) containing Krebs solution of the following composition (in mM): NaCl 136, KCl 5, CaCl_2_ 2.5, MgSO_4_ 1.2, KH_2_PO_4_ 1.2, NaHCO_3_ 23.8, and glucose 11, pH 7.5. The solution was aerated with a mixture of 95% O_2_ and 5% CO_2_ at 37 °C. Bipolar electrodes positioned in the region between the tendon and muscle were used to apply field stimulation with supramaximal electrical stimuli (frequency: 0.5 Hz; duration: 0.2 ms) delivered from a digital stimulator (model 100-C4, AVS Projetos). Muscle twitches were recorded via isometric transducers coupled to an amplifier and AQCAD software (AVS Projetos). The preparations were stabilized for 20 min prior to testing the fractions that were added in a fixed concentration of 10 µg/mL. The potential influence of RIPS and its fractions on postsynaptic cholinergic nicotinic receptors (indicative of neurotoxicity) and skeletal muscle fibers (indicative of myotoxicity) was assessed by comparing the contractile responses to 110 µM ACh and 24 mM KCl, respectively, before (basal responses) and after incubation with the secretion and fractions [90]. Muscle contractile responses were recorded for 120 min after the addition of RIPS or fractions.

#### 5.5.4. Acetylcholinesterase (AChE) Activity

The influence of RIPS fractions on AChE activity was assayed according to Ellman et al. [91] using chick brain slices and chick *biventer cervicis* muscle homogenates. Chick brain slices (a pool of eight slices incubated for 1 h with RIPS fraction) or each *biventer cervicis* muscle (collected at the end of the protocols described in the previous section) were homogenized (2000 rpm, 1 min) in 750 μL of phosphate-buffered saline (PBS; pH 7.0) using a bead-based homogenizer (Powerlyzer, MO BIO Laboratories, Inc., Carlsbad, CA, USA) and then centrifuged (1000× *g*, 5 min, 4 °C). A 50 μL aliquot of the resulting supernatant was mixed with 50 mM DTNB (5,50-dithiobis-(2-nitrobenzoic acid)) and the reaction was monitored at 405 nm in a Biotek ELx800 multi-well plate reader and analyzed with Gen5 Data Analysis software. Protein concentrations were determined using the Bradford dye-binding method [92]. The results were expressed as milliunits of AChE/mg protein (mU/mg protein), with one milliunit of activity defined as the amount of enzyme that produced 1 nmol of TNB (2-nitro-5-thiobenzoic acid)/min under the specified conditions.

### 5.6. Statistical Analysis

The data were expressed as the mean ± standard error of the mean (S.E.M.) and each experiment was repeated at least three times. Data from three or more experimental groups were analyzed by one-way ANOVA followed by Dunnett’s test (the groups were compared with a positive control or saline) or two-way ANOVA followed by the Tukey test, with a value of *p* < 0.05 indicating significance. All statistical analyses were done using GraphPad Prism 6.0 (Software Inc., San Diego, CA, USA).

## Figures and Tables

**Figure 1 toxins-12-00396-f001:**
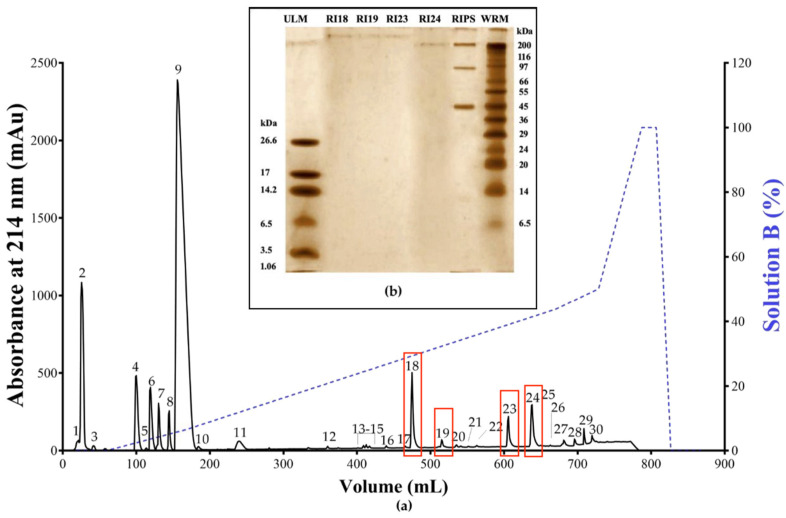
(**a**) Reversed-phase fast protein liquid chromatography (RP-FPLC) elution profile of *R. icterica* parotoid secretion (RIPS) and (**b**) Tricine-sodium dodecyl sulfate-polyacrylamide gel electrophoresis (SDS-PAGE) profile of fractions RI18, RI19, RI23, and RI24. RIPS (10 mg) was applied to a C18 column pre-equilibrated with 0.1% trifluoroacetic acid (TFA). The secretion components were then eluted (1 mL/min) with a two-step gradient of solution B (60% acetonitrile in 0.1% TFA) and the elution profile was monitored at 214 nm (solid line); the dotted line shows the gradient of solution B. Panel (**b**) shows the Tricine-SDS-PAGE profile of fractions RI18, RI19, RI23, and RI24 (2 µg each) in a 16.5% polyacrylamide gel stained with silver nitrate. The lanes on the extreme left and right show the ultra-low molecular mass markers (ULM) and wide-range molecular mass markers (WRM), respectively.

**Figure 2 toxins-12-00396-f002:**
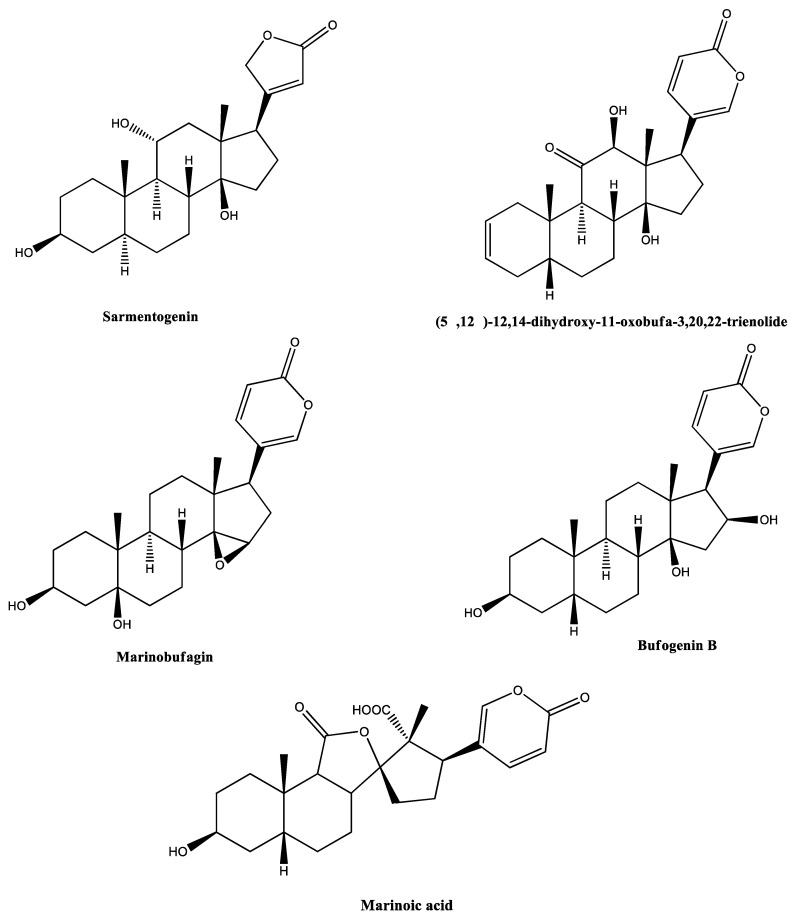
Chemical structures of the compounds identified by mass spectrometry in fractions RI18, RI19, RI23, and RI24 from *R. icterica* parotoid secretion. The structures were generated using Chemdraw professional software v.16.0.1.4 (PerkinElmer, Waltham, MA, USA).

**Figure 3 toxins-12-00396-f003:**
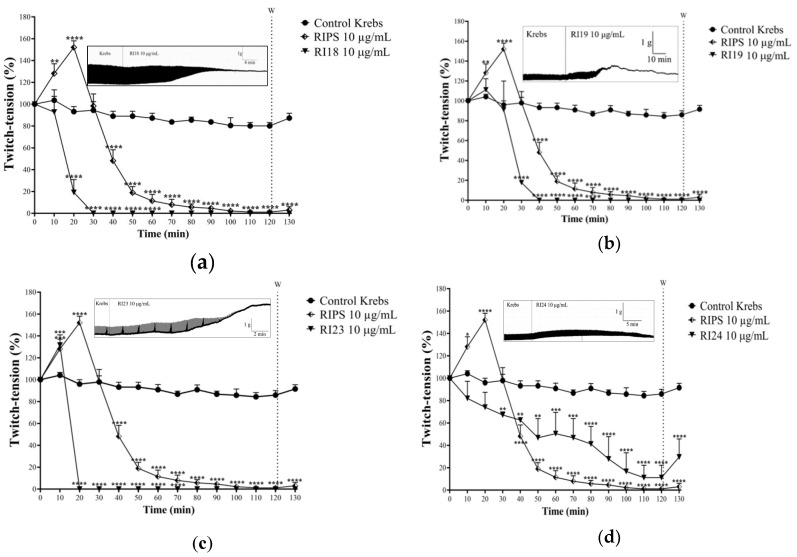
Neuromuscular effect of RIPS, fractions RI18, RI19, RI23, and RI24, and ouabain in chick *biventer cervicis* preparations. The panels show the neuromuscular responses to RI18 (**a**), RI19 (**b**), RI23 (**c**), RI24 (**d**), and 0.4 nM (1 µg/mL) ouabain (**e**). RIPS and the four fractions were all tested at a fixed concentration of 10 µg/mL. In all cases, the preparations were stimulated indirectly (0.5 Hz, 0.2 ms, 3–10 V) for 120 min. (**f**) Contractures to exogenous ACh (110 µM) and KCl (24 mM) in chick *biventer cervicis* preparations in the absence and presence of RIPS, the four fractions and ouabain. The results were expressed as the percent response relative to the basal tension (before addition of RIPS, fractions or ouabain) in panels (**a**–**e**) and to the basal responses to ACh and KCl in panel (**f**), and are shown as the mean ± standard error of the mean (S.E.M.) (*n* = 3 in all cases) * *p* < 0.05, ** *p* < 0.01, *** *p* < 0.001 and **** *p* < 0.0001 compared to the Krebs control in panels (**a**–**e**) and the Krebs control for ACh-induced contractures in panel (**f**); ^#^
*p* < 0.05 and ^####^
*p* < 0.0001 compared to the Krebs control for ACh-induced contractures in panel (**f**).

**Figure 4 toxins-12-00396-f004:**
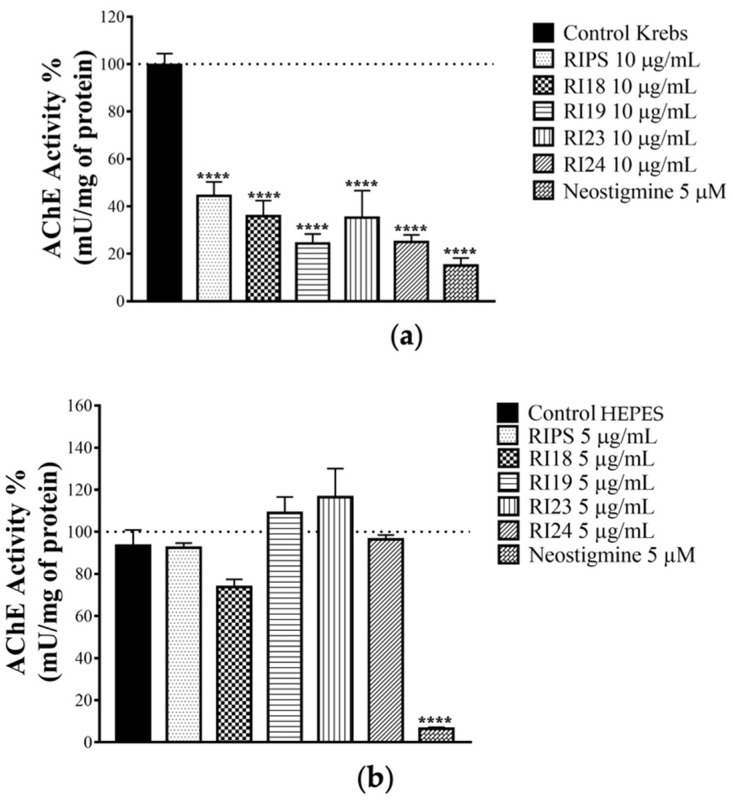
Modulation of acetylcholinesterase (AChE) activity by RIPS and its fractions. (**a**) AChE activity in homogenates of chick muscle from *biventer cervicis* preparations previously incubated with RIPS and its fractions (10 µg/mL each). (**b**) AChE activity in chick brain slices incubated with RIPS and its fractions (5 µg/mL each). In all experiments, 5 µM neostigmine and Krebs or HEPES were used as positive and negative controls, respectively. The columns represent the mean ± S.E.M. (*n* = 3). **** *p* < 0.0001 compared to HEPES or Krebs controls.

**Figure 5 toxins-12-00396-f005:**
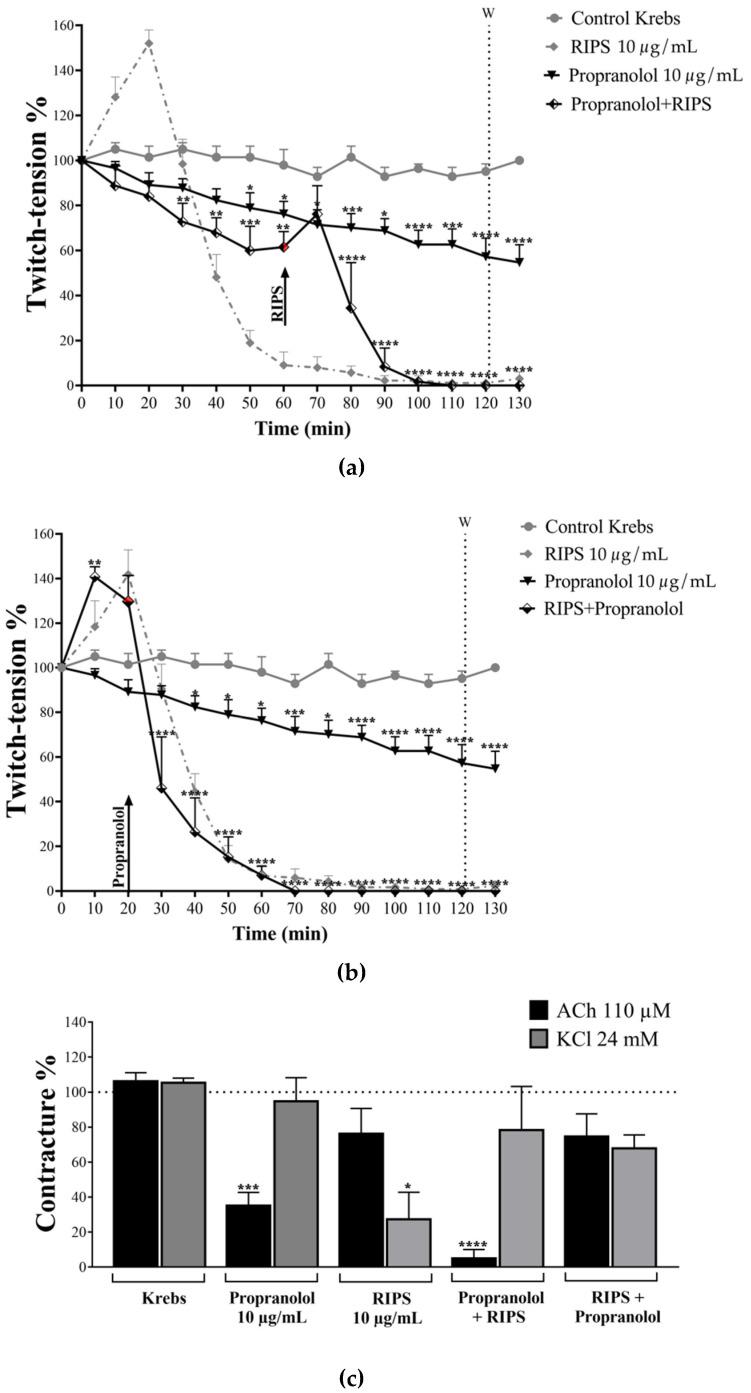
Effect of RIPS on propranolol-induced neuromuscular responses in chick *biventer cervicis* preparations. (**a**) Twitch-tension responses to propranolol (10 µg/mL, 38 µM) alone and to RIPS (10 µg/mL) after preincubation of the preparations with propranolol. (**b**) Twitch-tension responses to propranolol (10 µg/mL) in preparations preincubated with RIPS (10 µg/mL). (**c**) Responses to exogenous ACh (110 µM) and KCl (24 mM). In all protocols, the preparations were stimulated indirectly (0.5 Hz, 0.2 ms, 3–10 V) for 120 min after which they were washed (W in panels **a** and **b**) to assess the reversibility of the blockade (irreversible in all cases). The responses of negative (Krebs solution alone) and positive (RIPS alone) control preparations are also shown. The points and columns represent the mean ± S.E.M. (*n* = 4). * *p* < 0.05, ** *p* < 0.01, *** *p* < 0.001 and **** *p* < 000.1 compared to preparations incubated with Krebs solution alone.

**Figure 6 toxins-12-00396-f006:**
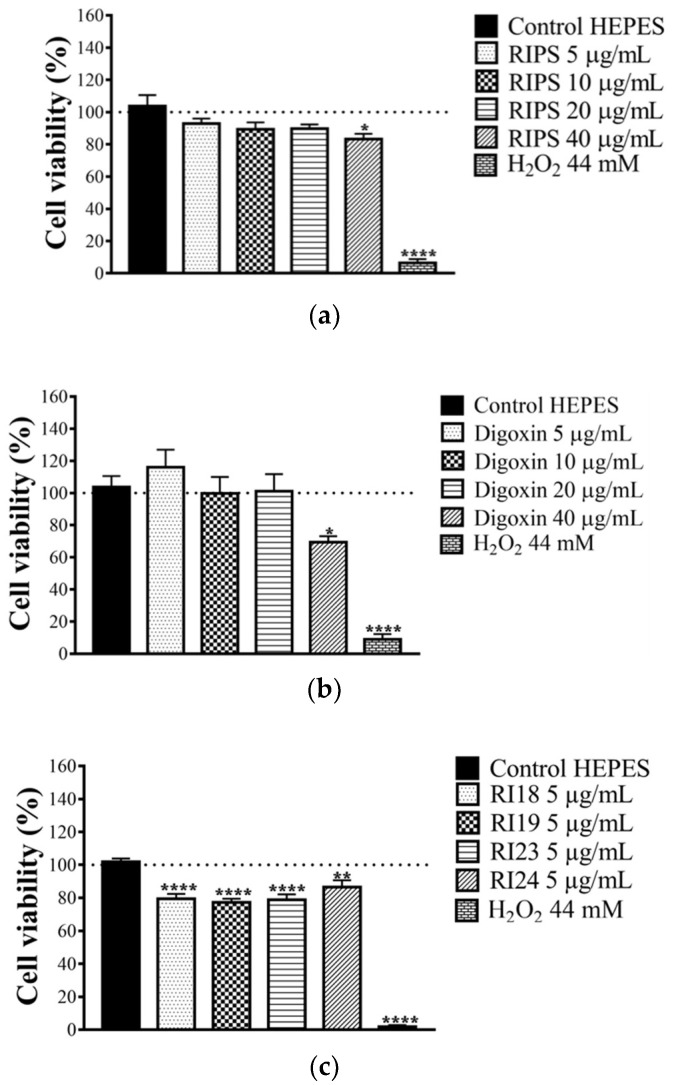
Effect of RIPS (**a**), digoxin (**b**), and RIPS fractions (**c**) on cell viability in chick brain slices. For the assays, brain slices were incubated with RIPS (5, 10, 20, and 40 µg/mL) and fractions RI18, RI19, RI23, and RI24 (all tested at a fixed concentration of 5 µg/mL) for 60 min prior to evaluating cell viability in the MTT (3-(4, 5-dimethylthiazol-2-yl)-2, 5-diphenyltetrazolium bromide) reduction assay. Digoxin (5, 10, 20, and 40 µg/mL), a Na^+^/K^+^-ATPase inhibitor, was included for comparison. The negative and positive controls were HEPES–saline buffer and 44 mM H_2_O_2_, respectively. The columns represent the mean ± S.E.M. (*n* = 6). * *p* <0.05, ** *p* < 0.01 and **** *p* < 0.0001 compared to the HEPES control.

**Table 1 toxins-12-00396-t001:** Chemical identification of the components of RIPS fractions by high resolution mass spectrometry, in positive and negative electrospray ionization modes (ESI(+) and ESI(−), respectively).

Entry	Precursor Ion *m/z*	Extract	Identification	Elem. Comp.	Diff ppm	Comp. Type	Ref.
Extract analysis in positive mode ESI(+)							
**1**	191.1171	RI24	*N*-Methyl-5-hydroxytryptamine	C_11_H_14_N_2_O	7.00	Alkaloid	[40,41,42,43]
**2**	391.2497	RI23	Sarmentogenin	C_23_H_34_O_5_	3.20	Steroid	[44]
**3**	399.2189	RI18	(5*β*,12*β*)-12,14-Dihydroxy-11-oxobufa-3,20,22-trienolide	C_24_H_30_O_5_	4.39	Steroid	[45,46]
	399.2182	RI23			2.63		
**4**	401.2306	RI18	14,15-Epoxy-3,5-dihydroxybufa-20,22-dienolide (marinobufagin)	C_24_H_32_O_5_	5.48	Steroid	[25,31,47,48]
	401.2318	RI23			2.50		
**5**	403.2446	RI23	(3*β*,5*β*,16*β*)-3,5,16-Trihydroxy-bufa-14,20,22-dienolide (Bufogenin B)	C_24_H_34_O_5_	9.55	Steroid	[49,50]
**6**	435.2356	RI19	11α,19-Di-hydroxy-telocinobufagin	C_24_H_34_O_7_	6.15	Steroid	[46,51,52]
	433.2266	RI24			9.17		
**7**	440.2167	RI23	Argentinogenin	C_26_H_31_O_6_	7.24	Steroid	[44,53]
**8**	445.2609	RI18	Bufotalin	C_26_H_36_O_6_	4.24	Steroid	[35,44,54,55,56,57]
**9**	458.2310	RI23	19-Oxo-cinobufagin	C_26_H_32_O_7_	1.19	Steroid	[36,44]
**10**	461.2481	RI24	Monohydroxylbufotalin	C_26_H_36_O_7_	8.73	Steroid	[6]
**11**	531.2983	RI23	3α,12*β*,25,26-Tetrahydroxy-7-oxo-5*β*-cholestane-26-*O*-sulfate	C_27_H_46_O_8_S	1.63	Steroid	[44]
**12**	599.3231	RI23	Cinobufagin-3-hemisuberate	C_34_H_46_O_9_	1.82	Steroid	[58]
**Extract analysis in negative mode ESI(−)**							
**13**	433.2266	RI24	11α,19-Di-hydroxy-telocinobufagin (Marinosin)	C_24_H_34_O_7_	9.17	Steroid	[46,52,53]
**14**	431.2116	RI19	Marinoic acid (3*β*-hydroxy-11,12-seco-5*β*,14*β*-bufa-20,22-dienolide-11,14-olide-12-oic acid)	C_24_H_30_O_7_	3.43	Steroid	[59,60]

Comp. type–component type (broad classification). Elem. comp.—elemental composition. RI–*Rhinella icterica.*

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
