# Peer review of "Chemical and Pharmacological Screening of Rhinella icterica (Spix 1824) Toad Parotoid Secretion in Avian Preparations"

_toxins, 2020, doi:10.3390/toxins12060396_

Round 1

Reviewer 1 Report

The manuscript presents a novelty on the Amphibian´s secretion field, with a pharmacological characterization of Rhinella icterica fractions on chick biventer cervicis neuromuscular preparations.

The experiments were well chosen and conducted to explain the effects.

Below some points that should be taken into account.

In the objectives, authors mentioned that they performed a ‘detailed investigation of the chemical composition of RIPS to identify the main toxic compound(s) responsible(s) for the acute poisoning caused by this secretion.’ I think this phrase should be rewritten as only one parameter of poisoning was investigated (neuromuscular activity) and fractions were studied instead pure molecules, thus no detailed chemical composition of the molecule(s) responsible for this effect could be attributed, but a set of them.

Moreover, in my opinion, the reproducibility of the crude secretion by a fraction is not the main goal, but the dissociation of two effects (e.g. facilitation of twitch-tension and neuromuscular blockade) after the fractionation, what shows that two molecules, responsible for different activities, could be separated and pharmacologically studied.

Regarding methods, some points that should be added in the manuscript:

  • How the toad´s secretion was obtained? By compression? And how do the authors determine 2 grams of such secretion?
  • The total amount of RIPS (5 mL) were injected in the FPLC in a single run?
  • The sentence that mention the FPLC gradient should be revised, once is confused if the gradient stops in the 60 or 100% (this information will be cleared in the results, in the figure that shows the fractioning).
  • The MTT assay was performed with 3 mm brain slices? The number of cells could be standardized for all groups (treated and control), once there is no control of how many cells are being added to the well (the calculation of cell viability percent can be mentioned). Moreover, using slices there is extracellular matrix and non-adherent cells that can give a false-negative result. Histological experiments of brain slices and analysis of cells could solve these problems.
  • How the chick biventer cervicis muscle homogenates were prepared for AChE assays?
  • The KCl and Ach contraction was induced after the chick biventer cervicis preparations?

In the results:

  • How the same molecule is present in fractions so distinct, like 19 and 24?
  • In the item 2.2 the authors commented that RIPS and fractions RI18, RI19, RI23 and RI24 were screened for neuromuscular activity in chick biventer cervicis preparations. However, it was mentioned before that this assay helped for fractions choosing. All the fractions were evaluated in the muscle preparation (as a screening) or did the authors choose 4 fractions based on their composition?
  • The figure 3e should contain the RI23 result without the inhibitor for a better comparison.
  • Why the experiments using propranolol were conducted only with RIPS and not with fractions? 

Reviewer 2 Report

The manuscript describes the chemical analysis of Rhinella icterica parotoid secretion (RIPS), as well as several biological activity studies performed with RIPS and its fractions. Chemical composition of RIPS and its fractions has been determined using mass spectrometry, and revealed a composition typical for toad parotoid secretion. The study of biological activities included: effect on neuromuscular activity, acetylcholinesterase activity and cell viability.

The topic of the manuscript is highly interesting, as the obtained results clearly indicate the complex mechanisms through which RIPS exerts its effect. While the obtained results clearly described and interpreted, the methodology requires some clarifications.

  • Neuromuscular blockade and propranolol-induced neuromuscular blockade have been determined at 10 μg/ml, while anti-AchE activity was determined at 5 and 10 μg/ml. Why not test neuromuscular blockade for 5 μg/ml as well?
  • Similarly, the propranolol-induced neuromuscular blockade has been studied using RIPS, but not its fraction. Explain this decision.
  • Please explain how the concentration values have been selected for the experiments.
  • Comparing identical concentration of RIPS and its fractions does not reflect the mixed composition of RIPS. The similar results of RI23 to RIPS could be related to quantity, not quality, e.g. if RIPS is mostly composed of RI23, their effect would obviously be similar. Furthermore, low quantity of other fractions in RIPS could explain the lack of response at the tested concentrations (but do not exclude an effect at higher concentrations). Reflect on the composition of RIPS compared to its fractions, when discussing the selected concentrations.
  • Lines 210-214: If all fractions caused significant decrease in cell viability at 5 μg/ml why did RIPS only affect cell viability at 40 μg/ml? This result suggests that the effect of fractions is not comparable with that of RIPS, which would undermine the use of identical concentrations of RIPS/fractions in the other experiments. Please consider carefully the overall conclusion of individual experiments with regard to the comparison of RIPS and its fractions.

Some recommendations that could improve the overall merit of the manuscript:

  • Lines 2-4: The proposed title is unusually long, a more concise title would fit better. The use of “peculiar” should be avoided, as it is ambiguous.
  • Line 26: The keyword “Vertebrates” is highly generic, does not reflect the content of the manuscript.
  • Lines 42-47: There are two phrases related to the composition of toad secretions. Please rephrase to clarify their meaning.
  • Lines 53-54: References 24 and 32 relate to scorpion venom, while the structure of the phrase suggest the use of toad venoms.
  • Line 69: “responsible” instead of “responsible(s)”.
  • Lines 70-74: This paragraph represents the conclusion of the study. Should be replaced with the corresponding hypothesis/objective.
  • Lines 78-79: Were all 30 subfractions tested on chick biventer cervicis? The phrase at lines 114-115 indicates that only peaks 18, 19, 23 and 24 were tested. Please clarify.
  • Figure 1b: Improve quality / modify contrast. ~14 kDa band is not visible.
  • Line 84: “R. icterica” should be written in italic style.
  • Table 1: A new column with component type (alkaloids, biogenic amines, steroids etc.) would be useful.
  • Table 1: If possible, enlarge width of "Identification" column, so that long names do not brake across rows (e.g. sarmentogenin).
  • Figure 2: Please specify the software used to generate the chemical structures or the source of these structures.
  • Line 112: “R. icterica” should be written in italic style.
  • Line 308-310: Elaborate the potential biotechnological application of individual compounds from RIPS in view of the suggested mechanism(s) of action.
  • Materials and Methods: Reference numbers should be in the order of appearance in text. References in Materials & Methods should follow those in other parts of the manuscript.
  • Line 365, 370, 470: Define FWHM, HEPES and PBS at first use.
  • References: Update the “References” list with more recently published articles to reflect the current relevance of the topic.

Reviewer 3 Report

This manuscript purports to document the chemical and pharmacological properties of toad (Rhinella icterica) parotoid secretion in avian preparations. These studies are carried out on crude parotoid secretion (RIPS), as well as four chromatographic fractions (RI18, RI19, RI23 and RI24), for which the chemical composition is claimed to be documented by mass spectrometric analysis. As knowledge of chemical composition is critical to the reproducibility and relevance of any pharmacology assessment/commentary, I was very disappointed to see that this study lacks the evidence to sustain any claims to chemical composition. The mass spectrometric data provided in Table 1 is incapable of providing unambiguous assignments of chemical structures. It is also worrying that although RI18 and RI23 are well resolved by HPLC (see Fig 1), both apparently contain the same metabolites 3 and 4. Likewise RI19 and R124 apparently contain 6. As I have no confidence that the authors have provided the analytical evidence critical to effective, reproducible and defendable chemical analysis of test samples, this seriously devalues the value of any conclusions.

Round 2

Reviewer 2 Report

Thank you to the authors for the detailed responses given and the modifications done to the manuscript.